# Digital Storytelling as an Intervention for Older Adults: A Scoping Review

**DOI:** 10.3390/ijerph20021344

**Published:** 2023-01-11

**Authors:** HeeKyung Chang, YoungJoo Do, JinYeong Ahn

**Affiliations:** 1College of Nursing, Gerontologic Health Research Center in Institute of Health Science, Gyeongsang National University, Jinju-si 52727, Gyeongsangnam-do, Republic of Korea; 2College of Nursing, Gyeongsang National University, Jinju-si 52727, Gyeongsangnam-do, Republic of Korea; 3Gerontologic Health Research Center in Institute of Health Science, Gyeongsang National University, Jinju-si 52727, Gyeongsangnam-do, Republic of Korea

**Keywords:** digital, storytelling, intergeneration, older adults, review

## Abstract

The population of older adults is rapidly increasing worldwide. Owing to fewer interactions between generations, older adults experience ageism and various psychological issues, such as depression and loneliness. Digital storytelling (DST) has the potential to share vivid lived experiences, support the forming of social relationships, and lead to improved well-being. This scoping review examines the potential psychosocial benefits of individual DST interventions for older adults and people with dementia. We adopted the methodological framework for scoping reviews outlined in the Joanna Briggs Institute’s (JBI) manual. A scoping review was performed using the following bibliographic databases: Web of Science, PubMed, Cochrane Library, CINAHL, Research Information Sharing Service, and National Assembly Library. There were 395 references retrieved, of which 19 articles were selected after applying inclusion and exclusion criteria. Our findings revealed that the most common effects of DST on older adults included the promotion of mental health, an increased amount of meaningful community connections, greater digital literacy, the mitigation of negative ageism, and enhanced intellectual ability. We suggest randomized controlled trials are conducted to confirm the efficacy of intergenerational DST intervention and the effects of DST interventions at multilevel outcomes, including the community level.

## 1. Introduction

Storytelling is a natural and universal form of communicating [1]. Historically, knowledge and skills have been bequeathed via word of mouth. As stories can be modified and added during delivery, the speaker’s and listener’s imagination is infused for future re-telling, thereby creating potentially different stories [2]. Digital storytelling (DST) is a creative way for people to share their stories that involves fusing digital media with voices, images, video, and music [3]. This allows for interactions across diverse populations, broadening the conversation window and increasing social connectivity for older adults whose scope of activities is gradually narrowing [4].

Older adult populations are rapidly increasing. In 2050, one out of four persons will be 60 years or older [5]. The expanding older population has generated a discourse about healthy aging. Healthy aging incorporates an individual’s physical and mental abilities in order to promote well-being in old age, and simultaneously seeks to develop and maintain their physical, social, and policy environments [6]. It requires a holistic and integrated approach that encourages creative expression, participation in social activities, lifelong learning, maintenance of individual abilities, disease prevention, and physical health promotion [7]. To ensure the healthy aging of older adults [7], it has been suggested that their quality of life should be improved; social networks and positive social interactions are such factors that can enhance quality of life. Thus, solving the social issues of older adults, such as social isolation and loneliness, is necessary for healthy aging [7,8]. Older adults can benefit from a positive digital world approach when tackling this societal issue [6]. For example, older adults’ use of information communication technology (ICT) devices positively impacted mental health and subjective well-being by reducing loneliness and increasing autonomy [9]. Positive utilization of ICT is reported to boost social engagement interaction and foster a sense of connection through contact with older people and generations [6,8,10]; thus, it can be effective in resolving societal issues such as social isolation and loneliness. Older adults’ access to digital technology expands their digital world and enhances their ability to utilize these devices and gain more digital literacy, which can improve their mental health and quality of life [8]. One concept for supporting healthy aging in older adults is DST activity [10].

In DST, older adults build a tale using their own language and expressions to express themselves utilizing DST [2,3,11]. Using multimedia technology, DST includes the fusing of pictures, audio, and narration to produce a film that represents one’s lived experiences. Older adults can become digital creators and develop their imagination and speech skills by sharing their stories with others [4]. By narrating stories to the younger generation, participating in DST, and expressing their identities and personalities, older adults with early-stage dementia were shown to have improved self-efficacy and depression symptoms [11]. Moreover, when older adults participated in DST, they maintained their memory, positively affecting their overall life. The digital literacy of older adults could be improved by using ICT to increase the quality of social interactions [3,11]. In a previous study, older adults who used tablet PCs could better operate electronic devices, had higher levels of self-efficacy and confidence, and had more energy due to their active participation in numerous tasks [12]. The use of broadly defined DST to enhance older adults’ health is a growing field of research.

Review studies confirmed the effects of personal reminiscence therapy [13] and autobiography [14] by limiting the participants to older adults with dementia; however, neither of the two studies [13,14] included digital technology. Rincon [15] performed a systematic review of DST studies but retrieving individual papers or verifying their subjects was challenging because the articles were integrated for each study purpose. Stargatt et al. [16] also conducted a systematic review of DST, but the results were limited as only autobiographical experiences and health-related outcomes were assessed. To know more about the DST intervention process and better understand its effects on older adults, it is necessary to first identify the concepts and structures needed in DST interventions and to generate a potential list of attributes before designing interventions for older adults. We confirmed that a scoping review is the best approach to answer these research questions as the knowledge needs that prompted the project are congruent with the kinds of outputs a scoping review produces [17,18]. This scoping review could guide the development of more effective DST interventions for older adults. Our primary objective for this review is to frame the structures that can be used for applications of DST and to identify the effects of DST intervention through evidence synthesis.

## 2. Materials and Methods

We conducted a scoping review following Joanna Briggs Institute (JBI)’s Manual for Scoping Reviews [19]. It comprises the following consecutive stages: identifying the research question; identifying relevant studies; study selection; charting the data; and collating, summarizing, and reporting results. The Preferred Reporting Items for Systematic Reviews and Meta-Analyses extension for scoping reviews (PRISMA-ScR) checklist guided this review [20]. The protocol for this scoping review has not been registered or published.

### 2.1. Research Questions

This review answers the following questions:
How is a DST intervention structured and applied to older adults?What are the effects of DST intervention on older adults?

### 2.2. Identifying Relevant Studies

We used the JBI reviewer’s manual and PRISMA-ScR guidelines to develop a comprehensive search strategy [19,20]. A broad systematic search was performed in December 2021 using the following databases: the Web of Science, PubMed, Cochrane Library, the Cumulative Index to Nursing and Allied Health (CINAHL), the Research Information Sharing Service, and the National Assembly Library.

The search terms were chosen to describe the population characteristics and the activities necessary for the review. The following syntax describes the terms that were used to search for the studies: (story*) AND (digital* OR ICT OR “artificial intelligence” OR online OR media OR mobile OR platform) AND (elder* OR older* OR dementia OR Alzheimer OR “cognitive impairment”) in Appendix A.

### 2.3. Study Selection

All articles in this review were peer-reviewed between the years 2002 and 2021. The inclusion and exclusion criteria (Table 1) were determined by conducting a review meeting regarding the selection of studies. Per the guidelines for reviewing the scope of the subject, two reviewers (the first author and corresponding author) examined 83 studies, including abstracts and texts, to improve the consistency of the search. Disagreements between the two researchers regarding study selection were resolved through additional discussions and agreements. In the event of any disputes, a third reviewer was to be consulted, but this was not required.

### 2.4. Data Charting

We extracted data from the included publications using a standardized data charting form while maintaining the terminology from the papers. The form included information on the authors, year of publication, country of origin, aim, purpose, population and sample size, DST intervention, design, method, recipients or observers, methods for sharing stories, and results (Table A1, Appendix B).

### 2.5. Collating, Summarizing, and Reporting the Results

We employed an inductive approach to thematically organize and summarize the results from the included papers to answer the research questions [21].

### 2.6. Quality of Included Articles

The Mixed Methods Appraisal Tool (MMAT) was used to assess the methodological quality of the included studies [22]. The MMAT is a critical evaluation tool created for the evaluation phase of systematic mixed study reviews of empirical research, or reviews that incorporate studies using mixed, qualitative, and quantitative methodological approaches. It comprises distinct sets of criteria to assess study validity in each of the study designs. Although it is recommended to report on each criterion scored in the appropriate section of the appraisal tool in order to provide a true value of the quality of each study, for this review, an overall score was calculated from a mean score of all items in the pertinent section of the checklist. Each “Yes”, “No”, and “Unclear/can’t tell” acquired a nominal value of 2, 0, and 1, respectively. Nevertheless, the authors concurred that the scoring enables the reader to consider the relative quality of the publications. There are no cut-off scores for the MMAT when identifying the quality of research. Notably, all papers are included and handled identically for the scoping review, regardless of quality in Appendix A.

## 3. Results

Database and manual searches yielded 395 publications. We screened the titles and abstracts of 259 publications after removing 105 duplicates. Based on the inclusion and exclusion criteria, the full text of 53 publications was read, 34 publications were excluded, and 19 were included in the review (see Figure 1).

### 3.1. Description of the Included Studies

Nineteen papers were included. The studies were conducted in Australia (*n* = 1), Brazil (*n* = 1), Canada (*n* = 5), Finland (*n* = 1), Germany (*n* = 1), Sweden (*n* = 1), Turkey (*n* = 1), the United Kingdom (*n* = 5), and the United States (*n* = 3). They employed qualitative (*n* = 12), mixed methods (*n* = 6), and quantitative (*n* = 1; randomized controlled trial) designs. Mixed methods studies mainly employed qualitative methods or open-ended questions post-intervention (*n* = 14), and quantitative studies involved questionnaires. Four, six, and nine studies were published between 2011 and 2013, 2014 and 2017, and 2018 and 2021, respectively.

Study quality was determined to be high and appropriate overall based on the MMAT’s quality assessment [22]. Two independent reviewers that contributed to the rating appraisal reached a consensus through debate.

### 3.2. Participants

The sample size of the included studies ranged from three to eighty-eight older adults; in six papers, the sample sizes were ten or fewer. The studies included patients with dementia (*n* = 5), patients with Alzheimer’s disease (*n* = 2), stroke survivors (*n* = 1), patients with an unreported diagnosis (*n* = 2), and patients with no diagnosis (*n* = 9).

The older adults engaged in DST across several age groups (50–99 years). Studies were conducted with elementary school-aged to college-aged students (*n* = 7), caregivers or trained volunteers (*n* = 3), spouses and/or families (*n* = 3), and other study facilitators. Appendix B presents the characteristics of the included studies.

### 3.3. Themes

1. How is a DST intervention structured and applied to older adults?

The results regarding application structure were framed as “connecting”, “mediating”, and “setting” in order to answer the research question on how storytelling intervention was performed using digital technology (see Table 2).

#### 3.3.1. Connecting

Individuals from various generations accompanied older adults in DST tasks. Older adults shared stories through DST with families, young people, and caregivers. After attending DST exhibitions, many community residents reported that they better understood and connected with the older adults [2,3,4,10,11,23,24,25,32,33,34].

#### 3.3.2. Mediating

The DST tool included pictures, audio, video, music applications, and GPS. The older adults and their partners engaged in DST using a tablet PC. For those unfamiliar with computers, DST was made easier through the use of a touchscreen device.

#### 3.3.3. Setting

The DST implementations adopted varied forms, such as websites, applications, or software. The stories were posted on social networking sites or made available on DVDs for participants and their families who wanted to view the stories after they were made. Workshops (*n* = 10), school processes (*n* = 3), and one-on-one sessions (*n* = 5) were used. Additionally, researchers visited seniors’ clubs and engaged in DST using various methods, such as teaching people.

### 3.4. Thematic Groupings

2. What are the effects of DST intervention on older adults?

The effects of DST intervention on older adults with or without mild cognitive impairment (MCI) were framed as follows (see Table 3).

#### 3.4.1. Promoting Older Adults’ Mental Health 

In 14 studies, older adults that experienced DST programs reported increased confidence, self-esteem, and self-efficacy, as well as reduced depressive symptoms. The older adults in these studies also reported improved mental health. The DST method was perceived as simple, effortless, and easy, thus enabling older adults to learn new things and undertake creative challenges that gave them a sense of pride and fulfillment [11,23,26,27,34]. DST allowed them to express themselves better than before, experience more pleasure, and be confident [4,10,11,12,23,24,26,34,35]. Moreover, they reported feeling livelier [4,25,29,30]. Subramaniam and Woods [24] stated that older adults reported lower depression following a DST intervention, and their caregivers and relatives corroborated this. Some older adults, for whom DST was applied with reminiscence therapy, had difficulty reviving unpleasant memories [24,26], but most reported improved self-confidence and well-being.

#### 3.4.2. Meaningful Community Connection

Thirteen studies reported that the DST intervention helped older adults feel more connected with others than before. They enjoyed sharing their stories, talking to others, and engaging in meaningful interactions [2,4,23]. They also participated in a digital life history project [2,3,10,25,30] to share their family roots and traditions with future generations. Students [2,3,4,10,23,25] and spouses [23,25,30,31,35] who accompanied older adults in these tasks, and the older adults themselves, reported a greater sense of belonging and strengthened relationships [2,3,4,10] after performing the tasks. The social associations among the participating students improved [25]. Sharing the completed stories on public platforms enabled them to understand each other better and have meaningful interactions about their experience [2,3,10,11,23,25,27,31,32,34,35]. Additionally, some participants felt that the shared experiences helped to create a social space [2,3,4,25,27,30]. Community partners, such as the local older adult housing community and historical commission, made the communities livelier than before.

#### 3.4.3. Achieving Digital Literacy

Per seven studies, older adults learned new things, became accustomed to digital technology, and gained more access to ICT. Older adults were more willing to learn new skills [3,10,12,26,35] when creating their own stories or expressing their lives by further thinking about them and re-imagining them [10,11,12,28,31,35]. This was achieved by turning still photographs into live pictures, thus adding the elements of sound and life [10,28,31]. Such activities with a creative focus can be particularly effective for people with dementia, as indicated by DST’s positive effects. During interviews before the project, older adults were worried about participating in the storytelling program due to unfamiliar and digital technology being used [11,23,26]. However, even though they operated the technology for the required time, they reported that they were happy to create stories [3,23,31]. By learning how to use computers and tablet PCs to shoot images or videos and becoming accustomed to using digital skills, older adults improved their level of digital literacy [3,23,31].

#### 3.4.4. Mitigating Negative Ageism

Five studies reported that young people’s (elementary to graduate students) perceptions of older adults changed positively, reducing their prejudices and discriminatory behaviors. In these studies, the students engaged in DST with older adults [12,23,25,32,33]. As participants, young people felt empathy toward older adults when they could relate to some of their experiences, thus forming a more positive perspective. It demonstrated that they were not simply cranky older adults, but were individuals with a wealth of knowledge and life experience who have much to impart to younger generations [33]. Older adults’ life experiences and social and work values deepened young adults’ understanding and reduced biases toward them [23].

#### 3.4.5. Enhancing Intellectual Ability

Across the six studies, DST improved memory and knowledge in older adults through the process of recalling their forgotten memories [11,24,26,27,30]. The participants recalled their past while thinking about the stories they could tell their children, nieces, or nephews [3,24,26,30]. Watching videos made by other older adults also enabled them to remember their past and revive additional memories [26]. Subramaniam and Woods [24] used the autobiographical memory interview extended version (AMI-E) and found that all participants’ knowledge scores improved. Although older adults and those with dementia had limited memory capacity, they could maintain cognitive function through repetitive learning [27]. Mental activity also helped maintain their skills and knowledge; however, the impact of dementia was evident in their memory loss and lethologica [26,27].

## 4. Discussion

This scoping review mapped and assessed published studies on DST among older adults and confirmed its application structure and effectiveness for them. This review summarizes the methods used to create digital stories, factors of digital story products, and outcomes and implementations of DST activities. The outputs were characterized as “connecting”, “mediating”, and “setting” in order to answer the research question concerning how digital technology has been structured to implement storytelling interventions. According to this structure frame, DST interventions utilizing various media and contexts might be developed by incorporating a range of participants. Although relatively simple, DST can stimulate an individual’s cognitive and sensory functions, resulting in various indirect experiences. The development of ICT has made it easy to create videos using a tablet PC or desktop application. DST can be used repeatedly, regardless of whether the older adults are community or nursing home residents. Furthermore, creating an online platform where older adults can share stories may encourage interactions with their family and community. Sharing life stories can create opportunities that allow members of the community to understand the experiences of others [3].

Engaging in DST was able to provide great vitality and prevent older adults living in nursing homes from feeling as though they were “dead weight” [4]. Moreover, they could experience improved connectedness and well-being through non-contact DST during the COVID-19 pandemic. Reciprocal story sharing can reduce feelings of isolation through the forming of social connections and new relations [36]. Although face-to-face interactions were reduced due to COVID-19, the non-contact method of using the platform could create a new way of communicating for them.

In this review, we found that the most common effects of DST on older adults included the promotion of mental health, an increased amount of meaningful community connections, greater digital literacy, the mitigation of negative ageism, and enhanced intellectual ability. Since the effects of DST intervention were identified at the individual and community levels, it is possible to further develop DST interventions and examine their effects across a range of outcomes.

The older adults and young people improved their understanding of one another through DST, thereby reducing ageism and improving young people’s perceptions of older adults [27,28,35]. Thus, DST can help improve relations between older and younger generations through meaningful interactions [33,35]. We further recommend randomized controlled trials using intergenerational DST interventions and systematic reviews to examine the community-level outcomes of DST interventions.

There were twelve qualitative studies and six mixed methods studies. Qualitative evaluations were conducted using semi-structured interviews and open-ended questions. Mixed methods studies quantitatively evaluated older adults with MCI and dementia, as well as their spouses, family members, and caregivers. DST was found to be helpful in managing older adults’ depression and quality of life [4,29]. The self-assessment results of the participants with cognitive impairment may be unreliable as patients may have difficulty in quantitative evaluation because their short-term memory is relatively poor. However, the increased frequency of participation in the intervention program, expressions of smiling faces, or increased food intake can be evaluated. The studies conducted a before-and-after evaluation for those caregivers or researchers who observed the participants. Alternatively, an objective effect may be derived if the intervention is filmed or evaluated [24,31].

There is no established framework for DST programs. It is up to the creator to depict the story they wish. Some programs were created by applying recall therapy, though participants could also talk about their daily lives and convey their thoughts. The participants reported similar outcomes even though they had different baseline levels of cognitive impairment. They discussed the following: recovery of cognitive ability, reduction of depressive symptoms, improved mental health, and harmony among generations. The “old adult” creates a story, but various outcomes may be achieved depending on the generation involved. The study design may be based on this principle. The existence of diversity limits the ability to conduct rigorous quantitative research within such a framework. By revealing the core concept and its theoretical and practical implications, factors predicated upon the results of various studies, developing a program based on this framework is necessary. 

### Limitations

There may be many articles where older adults use digital media to tell stories. However, several studies may have been excluded because of the limited scope of the literature search. This research is limited to English and Korean language publications. There may be other DST papers written in other languages that were not included in this review. 

## 5. Conclusions

This scoping review can contribute to the use of DST in improving the quality of life and mental health of older adults with or without cognitive impairment. DST can be used to easily create a digital story anywhere with tools that users prefer and songs, photos, and paintings that they like. It can be applied to older adults using easy-to-use devices, such as tablet PCs and laptops, without any significant preparation and has the advantage of generating DST even in non-face-to-face situations. The older adults who experienced DST intervention gained confidence in digital technology, increased their e-literacy, decreased negative emotions such as depression and loneliness, and experienced intergenerational interaction. It is anticipated that extending DST intervention to older adults will improve their quality of life, increase their energy, and support their mental health. Furthermore, if the story created is shared with the community and the younger generation, it will reduce the gap between generations and help them to understand each other better. It is necessary to develop intervention programs tailored to the varied needs of older adults with different health conditions; furthermore, interventions to enhance community integration must be developed soon. Family members or young people are encouraged to appropriately participate in DST interventions for older adults to facilitate access to digital technology.

## Figures and Tables

**Figure 1 ijerph-20-01344-f001:**
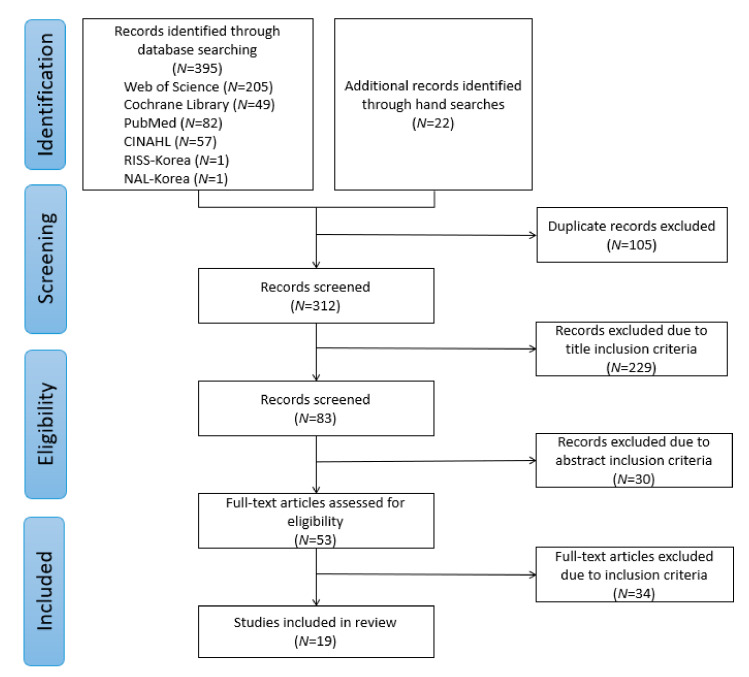
Flow diagram of study selection.

**Table 1 ijerph-20-01344-t001:** Inclusion and exclusion criteria.

Criterion	Inclusion	Exclusion
Type of studies	Qualitative, quantitative, and mixed methods studies published in peer-reviewed journals	Letters, comments, abstracts, editorials, doctoral theses, or any type of review
Period	From 1 January 2002, until 19 December 2021	
Language	English, Korean	
Type of participants	Older adults aged 50 years or older with mild cognitive impairment (MCI) or dementia	Cancer, patients with other diagnoses
Phenomenon of interest	Studies that include narrative-type media production that delivers stories using video, photo, audio, sound media, etc.	Narrative research without the use of digital media

**Table 2 ijerph-20-01344-t002:** Application structure for digital storytelling intervention.

Dimension	Components	Study	Number of Articles
Connecting with the older person’s companion	Older person	[3,4,10,11,23,24,25,26,27,28,29]	11
	Family	[2,3,10,11,23,25,27,29,30,31]	10
	Student (young person)	[2,3,4,10,12,23,25,32,33]	9
	Community member	[2,3,10,25,26,31,32,33,34]	9
	Caregiver	[11,24]	2
	Trained volunteer	[22,28]	2
Mediating DST tool	Picture	[2,3,10,11,12,23,24,26,27,28,29,30,31,32,33,34,35]	17
	Audio	[2,3,10,11,23,24,25,26,27,28,29,32,33,34]	14
	Video	[2,3,10,23,24,25,26,27,28,29,30,33,34]	13
	Music	[2,3,4,10,24,25,26,29,30,32,35]	11
	Application	[2,4,12,26,29,31,32]	7
	Touchscreen device	[26,31,35]	3
	GPS	[30]	1
Setting DST implementation	Software	[2,3,4,10,11,24,26,29,30,33]	10
	Workshop	[2,3,10,11,12,25,27,32,34]	10
	Website	[2,3,10,26,31,33,34,35]	8
	One-on-one session	[2,3,10,23,25,26,31,35]	8
	School	[12,23,25]	3
	Social network	[23,30]	2

**Table 3 ijerph-20-01344-t003:** Articles included in thematic groupings.

Theme	Study	Number of Articles
Promoting older adults’ mental health	[2,4,10,11,12,23,24,25,26,27,29,30,34,35]	14
Meaningful community connection	[2,3,4,10,11,23,25,27,30,31,32,34,35]	13
Achieving digital literacy	[3,10,11,12,28,31,35]	7
Mitigating negative ageism	[3,23,25,32,33]	5
Enhancing intellectual ability	[3,11,24,26,27,30]	6

## Data Availability

Not applicable.

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
