# Peer review of "Digital Storytelling as an Intervention for Older Adults: A Scoping Review"

_ijerph, 2023, doi:10.3390/ijerph20021344_

Round 1
Reviewer 1 Report
Congratulations for a comprehensive study, with a stringent topic.
On line 117 you mention a figure, but I could not find it.
However, when presenting the studies included in the analysis, you should rephrase the information regarding their characteristics, not simply copy them from the articles as this may be considered plagiarism.

Author Response
Response to Reviewer
Manuscript ID: ijerph- 2063459
We are pleased to be able to resubmit a revised version of manuscript No. IJERPH (ISSN 1660-4601), “Digital Storytelling as an Intervention for Older Adults: A Scoping Review” for publication in the International Journal of Environmental Research and Public Health. We have read the comments of the reviewers and have revised the manuscript. The manuscript has undoubtedly benefited from your comments and the reviewers’ insightful suggestions. We look forward to working with you and the reviewers to move this manuscript closer to publication in International Journal of Environmental Research and Public Health.
All reviewer comments have been addressed, with corresponding changes made directly to the manuscript where appropriate. To make the changes easier to identify where necessary, we have provided the corresponding line numbers.
- On line 117 you mention a figure, but I could not find it.
We apologize for the omission of figure 1, which has now been added.
- However, when presenting the studies included in the analysis, you should rephrase the information regarding their characteristics, not simply copy them from the articles as this may be considered plagiarism.
We have noted your advice and the appendix has been amended.

Reviewer 2 Report
Abstract
Please also add a concluding sentence after your result of the policy implications of this study at the end of the abstract.
Introduction
The introduction is very well written that cover the objective of the study. However, I would suggest emphasizing the importance and pieces of evidence on Digital Storytelling and mental well-being among the older population. Such evidence is important to build the argument on the importance of Digital Storytelling and the mental and overall health of older people and the scope of this review article.
I would also suggest mentioning the research questions at the end of the introduction section where the authors are building the narrative of the importance of this study.
Methods
Did you use the Scopus database for this study? It is not mentioned in the abstract. However, the authors mentioned the search strategy of Scopus in the method section.
Add a flow diagram on, how did the authors finalize the final papers. I think it is missing from the paper.
Provide the complete search strategy of each database as an Annexure.
Participants:
It is also important to mention the type of studies, e.g. qualitative or quantitative before referring to the sample size.
Some core themes of the studies are too long. I would suggest renaming the lengthy themes in a brief headline.
The conclusions section can be rewritten keeping in view the mentioned objective of the study. Do not mention the methodology of the paper in the conclusions. Put more emphasis on the implication of the study and the future direction of the work.
Limitations of the study
The authors can also measure the quality of the qualitative papers by using the COREQ a 32-item checklist written by Tong. What are the methodological limitations of this study such as search strategy etc?
Author Response
We are pleased to be able to resubmit a revised version of manuscript No. IJERPH (ISSN 1660-4601), “Digital Storytelling as an Intervention for Older Adults: A Scoping Review” for publication in the International Journal of Environmental Research and Public Health. We have read the comments of the reviewers and have revised the manuscript. The manuscript has undoubtedly benefited from your comments and the reviewers’ insightful suggestions. We look forward to working with you and the reviewers to move this manuscript closer to publication in International Journal of Environmental Research and Public Health.
All reviewer comments have been addressed, with corresponding changes made directly to the manuscript where appropriate. To make the changes easier to identify where necessary, we have provided the corresponding line numbers.
Comments of Reviewer #2:
Abstract
- Please also add a concluding sentence after your result of the policy implications of this study at the end of the abstract.
Thank you for your advice. I have added it in the last sentence of the abstract:
“We suggest randomized controlled trials to confirm the intergenerational DST intervention and the effects of DST interventions at multilevel outcomes including the community level.”
Introduction
- The introduction is very well written that cover the objective of the study. However, I would suggest emphasizing the importance and pieces of evidence on Digital Storytelling and mental well-being among the older population. Such evidence is important to build the argument on the importance of Digital Storytelling and the mental and overall health of older people and the scope of this review article.
Thank you for your suggestion. We have added it in the last sentence of the fourth paragraph (lines 60–70):
“Older adults can become digital creators and develop their imagination and speech skills by sharing their stories with others [4]. By narrating stories to the younger generation, conducting DST, and expressing their identities and personalities, older adults with early-stage dementia improved their self-efficacy and depression [11]. Moreover, when older adults conducted DST, they maintained their memory, positively affecting their overall life. The digital literacy of older adults could be improved by increasing the quality of social interaction using ICT [3,11]. Older adults who used tablet PCs could operate better electronic devices, had higher levels of self-efficacy and confidence, and had more energy due to their active participation in numerous tasks [12]. The use of broadly defined DST to enhance older adults’ health is a growing field of research.”
- I would also suggest mentioning the research questions at the end of the introduction section where the authors are building the narrative of the importance of this study.
Thank you for your suggestion. We have added it towards the end of the fourth paragraph (lines 77–81):
“To know more about the DST intervention process and better understand its effects on older adults, it is necessary to first identify the concepts and structure for applying DST intervention and generate a potential list of attributes before designing and developing the DST intervention for older adults.”
Methods
- Did you use the Scopus database for this study? It is not mentioned in the abstract. However, the authors mentioned the search strategy of Scopus in the method section. Add a flow diagram on, how did the authors finalize the final papers. I think it is missing from the paper. Provide the complete search strategy of each database as an Annexure.
We are grateful for your constructive comment, which allows us to interpret the results better. Scopus was referenced as a search engine in the text, although it was not used. The search process was changed based on the search engine. We added Figure 1 because it was missing.
Participants:
- It is also important to mention the type of studies, e.g. qualitative or quantitative before referring to the sample size.
Thank you for pointing this out. In “3.1 Description of the Included Studies,” we mentioned the type of studies (lines 161–162):
“They employed qualitative (n=12), mixed-methods (n=6), and quantitative (n=1; randomized controlled trial) designs”
- Some core themes of the studies are too long. I would suggest renaming the lengthy themes in a brief headline.
Thank you for your advice. We renamed the themes; they are now brief. The new brief core themes headlines are:
Promoting older adults’ mental health, Meaningful community connection, Achieving digital literacy, Mitigating negative ageism, Enhancing intellectual ability.
- The conclusions section can be rewritten keeping in view the mentioned objective of the study. Do not mention the methodology of the paper in the conclusions. Put more emphasis on the implication of the study and the future direction of the work.
We are grateful for your constructive comment. We have revised the conclusions section based on your recommendation (lines 331–347)
Limitations of the study
- The authors can also measure the quality of the qualitative papers by using the COREQ a 32-item checklist written by Tong. What are the methodological limitations of this study such as search strategy etc?
Thank you for your advice. We have considered your suggestion and employed the MMAT to conduct a quality assessment. This is discussed in section 2.6 Quality of Included Articles (line 136)

Reviewer 3 Report
Thank you very much for allowing me to read this interesting manuscript. I appreciate the interest of researchers in trying to review about Digital Story telling as an Intervention for Older Adults. Nevertheless, there are some issues with the manuscript that make the results less credible. Improvements are required to the current iteration before publication can be recommended. Please see the below comments and recommendations
Keywords are not appropriate to the content of the manuscript. Please try to adjust to the key terms.
Introduction.
The introduction is poor, it should be clear.
Line 36-40 / 49 -53. It is not well understood. Please rephrase the ideas.
Line 50 “early dementia”. what it refers to, can the authors provide background knowledge on this term.
Cite [13] [14] are systematic reviews. Why the authors have decided to do a Scoping Review on this topic when previous reviews exist. Consistent with the concept of scoping review, Scoping Review are useful fo answering much broader questions, not clearly defined questions.
Line 62-66: does not have a suitable connection with the rest of the text.
Line 67-68. What is it necessary for? The objective should be clearly defined at the end of the introduction
Materials and Methods.
According to PRISMA-ScR you should present the full electronic search strategy for at least one database, including any limits used, such that it could be repeated.
From line 90 onwards, the authors facilitate the search in SCOPUS, however, this platform is not listed among the databases used in the search.
Table 1. Exclusion criteria should not be seen as the opposite of inclusion criteria
Results
Searching in SCOPUS according to the proposed terms and on the suggested dates, the search yields a total of more than 600 studies, but you report 368 publications found.
Figure 1 is missing.
Discussion
The discussion is not well structured, it should start from an objective that is not well defined in the manuscript. The discussion as it is currently presented appears to be a way of reinforcing a single idea rather than discussing the results obtained with the current literature.
Limitations
Lines 274-275 “This was a scoping review and hence, there may have been a risk of methodological bias”: The Scoping Review methodology is extensively described in the literature, in different digital media and updated, so what does this assertion mean?
Author Response
We are pleased to be able to resubmit a revised version of manuscript No. IJERPH (ISSN 1660-4601), “Digital Storytelling as an Intervention for Older Adults: A Scoping Review” for publication in the International Journal of Environmental Research and Public Health. We have read the comments of the reviewers and have revised the manuscript. The manuscript has undoubtedly benefited from your comments and the reviewers’ insightful suggestions. We look forward to working with you and the reviewers to move this manuscript closer to publication in International Journal of Environmental Research and Public Health.
All reviewer comments have been addressed, with corresponding changes made directly to the manuscript where appropriate. To make the changes easier to identify where necessary, we have provided the corresponding line numbers.
Comments of Reviewer #3:
- Keywords are not appropriate to the content of the manuscript. Please try to adjust to the key terms.
Thank you for your advice. We have changed the keywords. The revised keywords are: digital; storytelling; intergeneration; older adults; review
Introduction.
- The introduction is poor, it should be clear.
Thank you for this comment. We have revised the introduction based on your suggestions.
2-1. Line 36-40 / 49-53. It is not well understood. Please rephrase the ideas.
We are grateful for this constructive comment. We corrected the second paragraph (lines 37–57):
“Older adult populations are rapidly increasing. In 2050, one out of four persons will be 60 years or older [5]. The expanding older population has generated a discourse about healthy aging. Healthy aging combines an individual’s physical and mental abilities that promote well-being in old age, alongside developing and maintaining their physical, social, and policy environments [6]. It requires a holistic and integrated approach that encourages creative expression, participation in social activities, lifelong learning, maintenance of individual abilities, disease prevention, and physical health promotion [7]. To ensure the healthy aging of older adults [7], it has been suggested that their quality of life should be improved; social networks and positive social interactions are such factors that enhance quality of life. Thus, solving the social issues of older adults, such as social isolation and loneliness, is necessary for healthy aging [7,8]. Older adults can benefit from the positive digital world approach to help tackle this societal issue [6]. For example, older adults’ use of information communication technology (ICT) devices positively impacted mental health and subjective well-being by reducing loneliness and increasing autonomy [9]. Positive technology utilizing ICT is reported to boost social engagement interaction and foster a sense of connection through contact with older people and generations [6,8,10]; thus, it can be effective in resolving societal issues like social isolation and loneliness. Older adults’ access to digital technology expands their digital world and enhances their ability to utilize these devices and gain more digital literacy, which can improve their mental health and quality of life [8]. One concept for supporting healthy ageing in older adults is DST activity [10].”
2-2. Line 50 “early dementia”. what it refers to, can the authors provide background knowledge on this term.
Thank you for your advice. This has been revised to “early-stage dementia.” (line 63).
Reference
Stenhouse, R.; Tait, J.; Hardy, P.; Sumner, T. Dangling conversations: Reflections on the process of creating digital stories during a workshop with people with early-stage dementia. J. Psychiatr. Ment. Health. Nurs. 2013, 20, 134–141. DOI:10.1111/j.1365-2850.2012.01900.x.
2.3 Cite [13] [14] are systematic reviews. Why the authors have decided to do a Scoping Review on this topic when previous reviews exist. Consistent with the concept of scoping review, Scoping Review are useful fo answering much broader questions, not clearly defined questions.
We are grateful for your constructive comment, which allows us to interpret the results better. We mentioned why we had to do a scoping review even though there is a systematic review related to DST as follows (lines 77–90):
“To know more about the DST intervention process and better understand its effects on older adults, it is necessary to first identify the concepts and structure for applying DST intervention and generate a potential list of attributes before designing and developing the DST intervention for older adults. We confirmed that a scoping review is the best approach to answer the research questions, ensuring that the knowledge needs that prompted the project are congruent with the kinds of outputs a scoping review produces; moreover, a systematic review is not a better option for this objective [17,18]. Our primary objective for this review is to frame the application structure and effects of DST intervention through evidence synthesis.
This list is not meant to be exhaustive; instead, it could guide the development of more effective DST interventions for older adults. Therefore, this scoping review aims to map the previous literature applying DST interventions to older adults and direct future research.”
- Line 62-66: does not have a suitable connection with the rest of the text.
We are grateful for this constructive comment. We have deleted these sentences.
2.5 Line 67-68. What is it necessary for? The objective should be clearly defined at the end of the introduction
We are grateful for this constructive comment, which allows us to interpret the results better. We have corrected the last paragraph (lines 87–90):
“This list is not meant to be exhaustive; instead, it could guide the development of more effective DST interventions for older adults. Therefore, this scoping review aims to map the previous literature applying DST interventions to older adults and direct future research.”
Materials and Methods.
- According to PRISMA-ScR you should present the full electronic search strategy for at least one database, including any limits used, such that it could be repeated.
We are grateful for your constructive comment. Accordingly, we have provided the following electronic search strategy for databases searched (lines 112–115):
“The following syntax describes the terms that were used to search for the studies: (story*) AND (digital* OR ICT OR “artificial intelligence” OR online OR media OR mobile OR platform) AND (elder* OR older* OR dementia OR Alzheimer OR “cognitive impairment”).”
- From line 90 onwards, the authors facilitate the search in SCOPUS, however, this platform is not listed among the databases used in the search.
Thank you for your advice. SCOPUS was referenced as a search engine in the text, although it was not used. The search process was changed based on the search engine.
- Table 1. Exclusion criteria should not be seen as the opposite of inclusion criteria
We are grateful for this constructive comment. We have modified the exclusion criteria.
Results
- Searching in SCOPUS according to the proposed terms and on the suggested dates, the search yields a total of more than 600 studies, but you report 368 publications found.
- Figure 1 is missing.
Thank you for your advice. SCOPUS was referenced as a search engine in the text, although it was not used. The search process was changed on the search engine. We have added Figure 1 to the manuscript.
Discussion
- The discussion is not well structured, it should start from an objective that is not well defined in the manuscript. The discussion as it is currently presented appears to be a way of reinforcing a single idea rather than discussing the results obtained with the current literature.
We are grateful for this constructive comment. We have revised the discussion section based on your recommendation.
Limitations
- Lines 274-275 “This was a scoping review and hence, there may have been a risk of methodological bias”: The Scoping Review methodology is extensively described in the literature, in different digital media and updated, so what does this assertion mean?
We are grateful for this constructive comment. We were concerned about the possibility of methodological bias since we failed to conduct a qualified evaluation. We have discussed another reviewer’s recommendation and used MMAT to conduct a quality assessment. This is described in section 2.6 Quality of Included Articles (line 136).

Round 2
Reviewer 2 Report
Thank you so much for addressing the comments and submitting the revised draft.
The authors have addressed the comments and have also revised the supplementary materials.
I am still a little bit confused about the search strategy used to carry out this study. For example, the author did not mention whether they searched the articles from "title search", "abstract", or the "whole article". This information is still missing.
Moreover, I would still suggest the search strategy mentioned in the headline "2.2. Identifying Relevant Studies" can further elaborate for better understanding.
Author Response
Second Response to Reviewer
Manuscript ID: ijerph- 2063459
We are pleased to be able to resubmit a revised version of manuscript No. IJERPH-2063459 (ISSN 1660-4601), “Digital Storytelling as an Intervention for Older Adults: A Scoping Review” for publication in the International Journal of Environmental Research and Public Health. We have read the comments of the reviewers and have revised the manuscript. The manuscript has undoubtedly benefited from your comments and the reviewers’ insightful suggestions. We look forward to working with you and the reviewers to move this manuscript closer to publication in the International Journal of Environmental Research and Public Health.
All reviewer comments have been addressed, with corresponding changes made directly to the manuscript where appropriate. To make the changes easier to identify where necessary, we have provided the corresponding line numbers.
Comments of Reviewer #2:
- I am still a little bit confused about the search strategy used to carry out this study. For example, the author did not mention whether they searched the articles from "title search", "abstract", or the "whole article". This information is still missing.
Moreover, I would still suggest the search strategy mentioned in the headline "2.2. Identifying Relevant Studies" can further elaborate for better understanding.
Response: Thank you for your advice. We have added the required information to the supplementary material for search string per Database.

Reviewer 3 Report
Although the authors have satisfactorily answered most of my questions, they should consider adding a few more changes.
Thank you for clarifying the concept and the reference to "early stage of dementia", however the research questions focus on the concept of "Mild cognitive impairment" so, in order to be consistent with your study, the introduction should explore this concept and its relevant references or modify the study questions.
Thank you for clarifying the reason for using scoping review as a review methodology, but this paragraph is not clear and the objetive/objectives are confusing and repetitive..
“To know more about the DST intervention process and better understand its effects on older adults, it is necessary to first identify the concepts and structure for applying DST intervention and generate a potential list of attributes before designing and developing the DST intervention for older adults. We confirmed that a scoping review is the best approach to answer the research questions, ensuring that the knowledge needs that prompted the project are congruent with the kinds of outputs a scoping review produces; moreover, a systematic review is not a better option for this objective [17,18]. Our primary objective for this review is to frame the application structure and effects of DST intervention through evidence synthesis.
This list is not meant to be exhaustive; instead, it could guide the development of more effective DST interventions for older adults. Therefore, this scoping review aims to map the previous literature applying DST interventions to older adults and direct future research.”
According to PRISMA-ScR "present the full electronic search strategy for at least 1 databases, including any limits used, such it could be reapeates, so that it can be repeated". The current version of the manuscript presents the syntax, but does not specify in which database.
“The following syntax describes the terms that were used to search for the studies: (story*) AND (digital* OR ICT OR “artificial intelligence” OR online OR media OR mobile OR platform) AND (elder* OR older* OR dementia OR Alzheimer OR “cognitive impairment”).”
Author Response
Second Response to Reviewer
Manuscript ID: ijerph- 2063459
We are pleased to be able to resubmit a revised version of manuscript No. IJERPH-2063459 (ISSN 1660-4601), “Digital Storytelling as an Intervention for Older Adults: A Scoping Review” for publication in the International Journal of Environmental Research and Public Health. We have read the comments of the reviewers and have revised the manuscript. The manuscript has undoubtedly benefited from your comments and the reviewers’ insightful suggestions. We look forward to working with you and the reviewers to move this manuscript closer to publication in the International Journal of Environmental Research and Public Health.
All reviewer comments have been addressed, with corresponding changes made directly to the manuscript where appropriate. To make the changes easier to identify where necessary, we have provided the corresponding line numbers.
Comments of Reviewer #3:
- Thank you for clarifying the concept and the reference to "early stage of dementia", however the research questions focus on the concept of "Mild cognitive impairment" so, in order to be consistent with your study, the introduction should explore this concept and its relevant references or modify the study questions.
Response: We are grateful for your constructive comment, which allows us to interpret the results of our study questions better. Our study participants included older adults who have not been diagnosed with dementia and survivors of mild to moderate dementia and stroke. Hence, we revised the research questions to:
- How is a DST intervention structured and applied to older adults?
- What are the effects of DST intervention on older adults?
As it is possible that a preceding study only mentioned participants with geriatric diseases, such as MCI, dementia, or AD, without mentioning the word elderly, these three were also included in the search terms.
- Thank you for clarifying the reason for using scoping review as a review methodology, but this paragraph is not clear and the objetive/objectives are confusing and repetitive..
“To know more about the DST intervention process and better understand its effects on older adults, it is necessary to first identify the concepts and structure for applying DST intervention and generate a potential list of attributes before designing and developing the DST intervention for older adults. We confirmed that a scoping review is the best approach to answer the research questions, ensuring that the knowledge needs that prompted the project are congruent with the kinds of outputs a scoping review produces; moreover, a systematic review is not a better option for this objective [17,18]. Our primary objective for this review is to frame the application structure and effects of DST intervention through evidence synthesis.
“This list is not meant to be exhaustive; instead, it could guide the development of more effective DST interventions for older adults. Therefore, this scoping review aims to map the previous literature applying DST interventions to older adults and direct future research.”
Response: Thank you for your advice. We have revised this section to be more concise and deleted any repeated information. Furthermore, we have added the information about this study being a scoping review toward the end of the fifth paragraph (lines 76–85):
“To know more about the DST intervention process and better understand its effects on older adults, it is necessary to first identify the concepts and structure for applying DST intervention and to generate a potential list of attributes before designing for older adults. We confirmed that a scoping review is the best approach to answer the research questions, ensuring that the knowledge needs that prompted the project are congruent with the kinds of outputs a scoping review produces [17,18]. This scoping reviews could guide the development of more effective DST interventions for older adults. Our primary objective is to frame the application structure and effects of DST intervention through evidence synthesis.”
- According to PRISMA-ScR "present the full electronic search strategy for at least 1 databases, including any limits used, such it could be reapeates, so that it can be repeated". The current version of the manuscript presents the syntax, but does not specify in which database.
“The following syntax describes the terms that were used to search for the studies: (story*) AND (digital* OR ICT OR “artificial intelligence” OR online OR media OR mobile OR platform) AND (elder* OR older* OR dementia OR Alzheimer OR “cognitive impairment”).”
Response: Thank you for your advice. We have added our electronic search strategy as supplementary material presenting the search string per Database.
